# A spontaneous genetically induced epiallele at a retrotransposon shapes host genome function

Tessa M Bertozzi[†], Nozomi Takahashi, Geula Hanin, Anastasiya Kazachenka[‡], Anne C Ferguson-Smith*

Department of Genetics, University of Cambridge, Cambridge, United Kingdom

**Abstract** Intracisternal A-particles (IAPs) are endogenous retroviruses (ERVs) responsible for most insertional mutations in the mouse. Full-length IAPs harbour genes flanked by long terminal repeats (LTRs). Here, we identify a solo LTR IAP variant (*Iap5-1^solo^*) recently formed in the inbred C57BL/6J mouse strain. In contrast to the C57BL/6J full-length IAP at this locus (*Iap5-1^full^*), *Iap5-1^solo^* lacks DNA methylation and H3K9 trimethylation. The distinct DNA methylation levels between the two alleles are established during preimplantation development, likely due to loss of KRAB zinc finger protein binding at the *Iap5-1^solo^* variant. *Iap5-1^solo^* methylation increases and becomes more variable in a hybrid genetic background yet is unresponsive to maternal dietary methyl supplementation. Differential epigenetic modification of the two variants is associated with metabolic differences and tissue-specific changes in adjacent gene expression. Our characterisation of *Iap5-1* as a genetically induced epiallele with functional consequences establishes a new model to study transposable element repression and host-element co-evolution.

*For correspondence:
afsmith@gen.cam.ac.uk

Present address: [†]Whitehead Institute, Cambridge, United States; [‡]The Francis Crick Institute, London, United Kingdom

Competing interests: The authors declare that no competing interests exist.

## Introduction

More than 10% of the mouse genome is made up of endogenous retroviruses (ERVs) (*Smit et al., 2015*). ERVs are transposable elements (TEs) containing retrotransposition-enabling genes flanked by non-coding identical 5′ and 3′ long terminal repeats (LTRs). Although most ERVs have lost their mobilisation potential due to mutational decay, they retain the ability to modulate host genome function through the use of transcriptional regulation motifs contained in their sequences. These include transcription factor binding sites, polyadenylation signals, and splice acceptor sites (*Maksakova et al., 2006*). For instance, solo LTRs produced via inter-LTR homologous recombination make up a large fraction of mammalian ERV sequences and can influence host gene expression through their promoter activity (*Nellåker et al., 2012*; *Ruda et al., 2004*; *Subramanian et al., 2011*).

Mammals have evolved a range of mechanisms to mitigate the deleterious effects of ERV retrotransposition and transcriptional disruption. The vast majority of mouse ERVs exhibit high levels of DNA methylation, and loss of DNA methyltransferase activity causes an increase in ERV transcription in embryos and in the germline (*Barau et al., 2016*; *Bourc'his and Bestor, 2004*; *Jain et al., 2017*; *Walsh et al., 1998*). In addition, the deposition of histone H3 lysine nine trimethylation (H3K9me3) by the methyltransferase SETDB1 (or ESET) in early development is crucial for ERV repression (*Matsui et al., 2010*). SETDB1 is recruited to ERVs following the binding of KAP1 (or TRIM28) to KRAB zinc finger proteins (KZFPs) that recognise specific sequence motifs within ERV elements (*Ecco et al., 2017*). RNA-mediated mechanisms such as the piRNA pathway also play key roles in silencing mammalian ERVs, especially in the male germline (*Aravin et al., 2007*; *Carmell et al., 2007*).

**eLife digest** Our genome provides a complete set of genetic instructions for life. It begins by directing the growth and development of the embryo, and subsequently supports all the cells of the adult body in their daily routines. Yet approximately 10% of the DNA in mammalian genomes is made up of sequences originating from past retroviral infections, leaving a calling card in our genetic code.

While these segments of retroviral DNA can no longer produce new infectious viruses, some of them retain the ability to copy themselves and jump into new parts of the genome. This can be problematic if they jump into and disrupt an important piece of genetic code. To protect against this, our bodies have evolved the ability to chemically strap down retroviral sequences by adding methyl groups to them and by modifying the proteins they are wrapped around. However, some of these endogenous retroviruses can dodge such so-called epigenetic modifications and disrupt genome function as a result.

Studying a population of widely used inbred laboratory mice, Bertozzi et al. have identified a retroviral element that evades these epigenetic restraints. They discovered that some mice carry a full-length retroviral sequence while others have a shortened version of the same element. The shorter sequence lacked the repressive epigenetic marks found on the longer version, and this affected the expression of nearby genes. Moreover, the repressive marks could be partially restored by breeding the short-version mice with a distantly related mouse strain.

Bertozzi et al. highlight an important issue for research using mouse models. Inbred laboratory mouse strains are assumed to have a fixed genetic code which allows scientists to conclude that any observed differences in their experiments are not a product of background genetic variation. However, this study emphasizes that this assumption is not guaranteed, and that hidden genetic diversity may be present in ostensibly genetically identical mice, with important implications for experimental outcomes.

In addition, Bertozzi et al. provide a new mouse model for researchers to study the evolution and regulation of retroviral sequences and the impact of these processes on cell function.

Intracisternal A-particles (IAPs) are murine-specific ERVs. They are young and highly active elements, exhibiting extensive insertional polymorphism across inbred mouse strains and accounting for more than 10,000 ERVs in the C57BL/6J (B6) inbred strain (*Nellåker et al., 2012*; *Smit et al., 2015*). IAP insertions are responsible for the majority of documented insertional mutations in the mouse, most of which disrupt host gene function by generating aberrant or fusion gene transcripts (*Gagnier et al., 2019*). In some cases, the phenotypic severity of the mutation is associated with the methylation status of the IAP. For example, the IAP LTR promoter at the *Agouti viable yellow* ($A^{vy}$) allele drives ectopic expression of the downstream coat colour gene *Agouti* when unmethylated. By mechanisms not yet fully understood, the $A^{vy}$ IAP exhibits variable DNA methylation levels across genetically identical individuals, which results in inbred mice displaying a range of coat colours (*Duhl et al., 1994*; *Morgan et al., 1999*).

We previously carried out a genome-wide screen to probe the generalisability of inter-individual methylation variability at IAPs (*Elmer et al., 2021*; *Kazachenka et al., 2018*). We identified dozens of novel variably methylated IAPs (VM-IAPs) in the B6 genome and showed that their characteristic methylation variability is recapitulated from generation to generation irrespective of parental methylation level. Our screen for VM-IAPs identified the IAP-*Pgm2* element, named after its closest annotated coding gene *Phosphoglucomutase-2* (*Pgm2*). It is a fully structured 7.5 kb IAP located on Chromosome 5 containing identical 5′ and 3′ LTRs of the IAPLTR2_Mm subclass (mm10 coordinates: chr5:64,030,834–64,038,297; *Figure 1—figure supplement 1*).

Here we show that IAP-*Pgm2*, in sharp contrast to VM-IAPs, exhibits two distinct DNA methylation states which are stably inherited from parent to offspring within the B6 strain. Sequencing of the locus reveals that the epiallele is genetically conferred, where the methylated variant of IAP-*Pgm2* is a full-length IAP matching the B6 reference genome (renamed *Iap5-1^full^*) and the unmethylated variant is a solo LTR produced from an inter-LTR recombination event (renamed *Iap5-1^solo^*). The absence of DNA methylation at *Iap5-1^solo^* is accompanied by a loss of H3K9me3 marks. We find

that differential modification of the two variants is established during early preimplantation development and identify candidate KZFPs responsible for the acquisition of contrasting epigenetic states. We report that *Iap5-1^solo* is unresponsive to dietary methyl supplementation but highly susceptible to genetic background, becoming a bona fide VM-IAP in an F1 hybrid context. In addition, we demonstrate that formation of the *Iap5-1^solo* allele is associated with tissue-specific changes in neighbouring gene expression and a decrease in fasting plasma glucose and triglyceride concentrations. Our study establishes the *Iap5-1* locus as a naturally occurring and biologically relevant model in the widely studied reference mouse strain to investigate the mechanisms underlying TE repression, inter-individual methylation variability, and TE-induced disruptions to host genome function.

## Results

### IAP-*Pgm2* methylation is tri-modally distributed in inbred B6 mice

Following our genome-wide screen for VM-IAPs in the B6 genome, DNA methylation at the distal CpGs of the 5′ LTR of each candidate was validated using genomic DNA (gDNA) extracted from adult inbred B6 mice (*Elmer et al., 2021*; *Kazachenka et al., 2018*). While VM-IAP DNA methylation levels display continuous probability distributions in the B6 population, the IAP-*Pgm2* 5′ LTR exhibited three distinct states: high (>85%), low (<20%), and intermediate (60–70%) methylation (*Figure 1A and B*). This pattern was observed in both sexes (*Figure 1—figure supplement 2*). In addition, 5′ and 3′ LTR methylation levels were consistent with one another within an individual (*Figure 1A and B*).

Methylation quantification of unique non-repetitive DNA immediately up- and downstream of IAP-*Pgm2* showed that the three distinct methylation states become less defined as the distance from the LTR borders increases, ultimately collapsing approximately 500 bp from either side of the IAP (*Figure 1C*). This provides evidence for short-distance spreading of DNA methylation levels from IAP-*Pgm2* into bordering DNA and suggests that the methylation differences observed between individuals are intrinsic to the IAP-*Pgm2* element rather than a reflection of differential methylation of the insertion site prior to integration.

### IAP-*Pgm2* methylation exhibits stable Mendelian inheritance

One of the characteristic properties of VM-IAPs is the reconstruction of inter-individual methylation variability from one generation to another regardless of parental methylation level (*Kazachenka et al., 2018*). To test whether this phenomenon occurs at IAP-*Pgm2*, specific parental combinations were set up for breeding and IAP-*Pgm2* 5′ LTR methylation levels were quantified in the offspring. In stark contrast to VM-IAPs, IAP-*Pgm2* exhibited stable inheritance of methylation levels. Offspring born to highly methylated parents were all highly methylated and offspring born to lowly methylated parents were all lowly methylated (*Figure 1D*). When one parent was highly methylated and the other lowly methylated, all offspring were intermediately methylated (*Figure 1D*). This indicates that high and low methylation states are allelic variants of IAP-*Pgm2* (designated IAP-*Pgm2*^HH and IAP-*Pgm2*^LL), with intermediate methylation representing co-dominant epigenetic heterozygosity (IAP-*Pgm2*^HL). Additional crosses confirmed this inheritance pattern: an IAP-*Pgm2*^HL intercross produced IAP-*Pgm2*^HH, IAP-*Pgm2*^LL, and IAP-*Pgm2*^HL offspring; an IAP-*Pgm2*^HH x IAP-*Pgm2*^HL cross produced IAP-*Pgm2*^HH and IAP-*Pgm2*^HL offspring; and an IAP-*Pgm2*^LL x IAP-*Pgm2*^HL cross produced IAP-*Pgm2*^LL and IAP-*Pgm2*^HL offspring (*Figure 1D*). These results additionally demonstrate that the methylation state of one IAP-*Pgm2* variant does not influence the methylation state of the other in a heterozygous context.

### The methylation state of IAP-*Pgm2* is genetically determined

The stable inheritance of IAP-*Pgm2* methylation is indicative of a spontaneous genetic mutation in the B6 population, either in the IAP element itself or in a gene involved in its epigenetic regulation. To investigate the former possibility, we designed PCR primers amplifying the entirety of IAP-*Pgm2* from IAP-*Pgm2*^HH and IAP-*Pgm2*^LL gDNA (*Figure 2A*). Nested primer pairs N1 and N2 amplified two overlapping fragments, each containing half of the IAP-*Pgm2* element (*Figure 2A*). Agarose gel electrophoresis of the PCR products revealed amplification of both fragments from IAP-*Pgm2*^HH

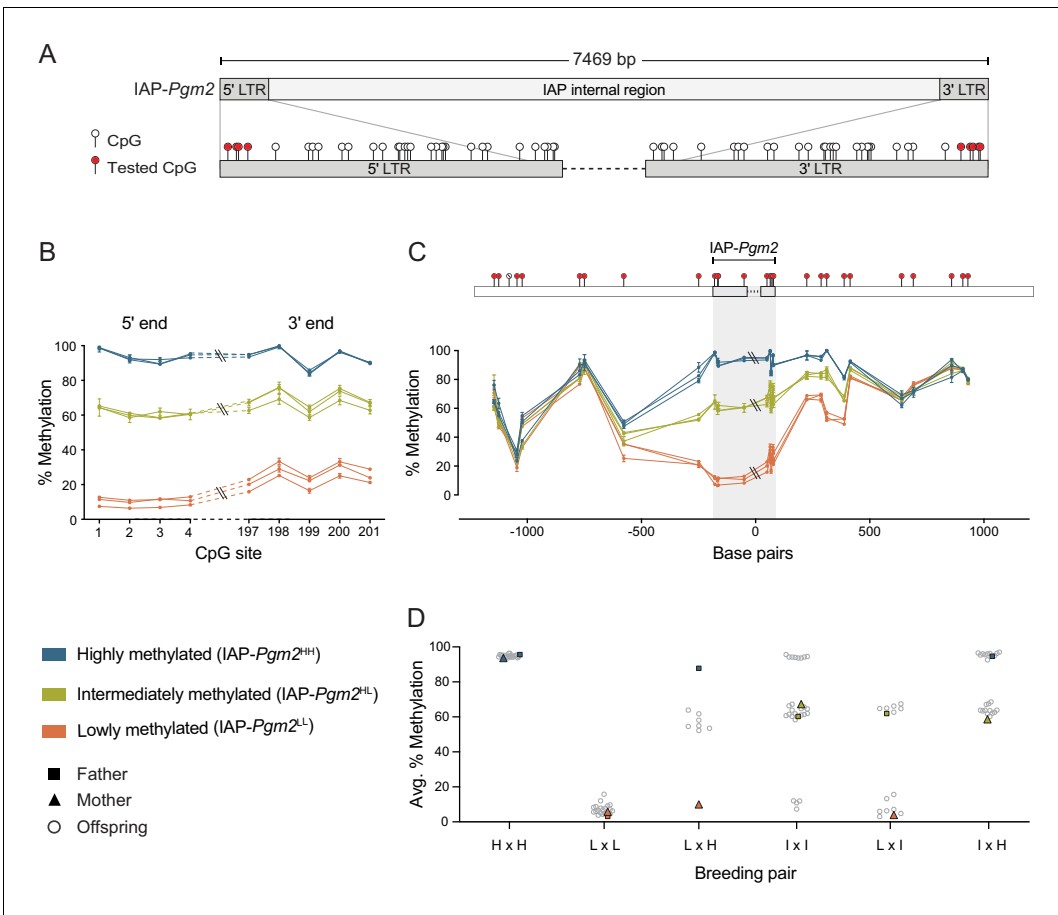

**Figure 1.** Intracisternal A-particle (IAP)-*Pgm2* methylation is tri-modally distributed and stably inherited in inbred B6 mice. (**A**) Map of CpG positions in the IAP-*Pgm2* long terminal repeats (LTRs). CpGs assayed in panel **B** are shown in red. (**B**) IAP-*Pgm2* methylation levels are consistent between the 5′ and 3′ ends. Methylation levels were quantified at the most distal CpGs of the IAP-*Pgm2* 5′ and 3′ LTRs (nearest to the boundary with unique DNA) using bisulphite pyrosequencing in ear DNA. Each point represents a CpG, each line represents an individual, and error bars represent standard deviations of technical triplicates. (**C**) Inter-individual methylation variation collapses within 500 bp on either side of IAP-*Pgm2*. Data presentation as in panel **A**. Assayed CpGs are shown in red above the graph. (**D**) Stable Mendelian inheritance of IAP-*Pgm2* methylation reveals that high (blue, H) and low (orange, L) methylation reflect two allelic states of IAP-*Pgm2*, with intermediate methylation (orange, I) representing heterozygosity. Each data point represents average methylation of ear DNA across the four most distal CpGs of the 5′ LTR for one individual.

The online version of this article includes the following source data and figure supplement(s) for figure 1:

**Figure supplement 1.** Genomic structure and coding potential of intracisternal A-particle (IAP)-*Pgm2*.

**Figure supplement 2.** The intracisternal A-particle (IAP)-*Pgm2* methylation pattern is not sex-linked.

**Figure supplement 2—source data 1.** Numerical data represented in panels B-D, and *Figure 1—figure supplement 2*.

gDNA but no amplification of either fragment from IAP-*Pgm2*^LL gDNA (*Figure 2B and C*), pointing to a substantial genetic difference between IAP-*Pgm2*^HH and IAP-*Pgm2*^LL individuals.

The nested primer pair N3 was designed to target the unique bordering regions on either side of the IAP element, amplifying an 8.5 kb fragment based on the GRC38/mm10 (mm10) genome assembly (*Figure 2A*). While no DNA bands were observed following the use of this primer pair on IAP-*Pgm2*^HH gDNA, a 1.5 kb band was amplified from both IAP-*Pgm2*^LL and IAP-*Pgm2*^HL gDNA (*Figure 2D*). The smaller amplicon is indicative of a 7 kb deletion on the IAP-*Pgm2*^L allele between the two primer annealing sites, while the lack of amplification from the IAP-*Pgm2*^H allele is likely due to the technical challenges associated with amplifying a large repetitive DNA fragment.

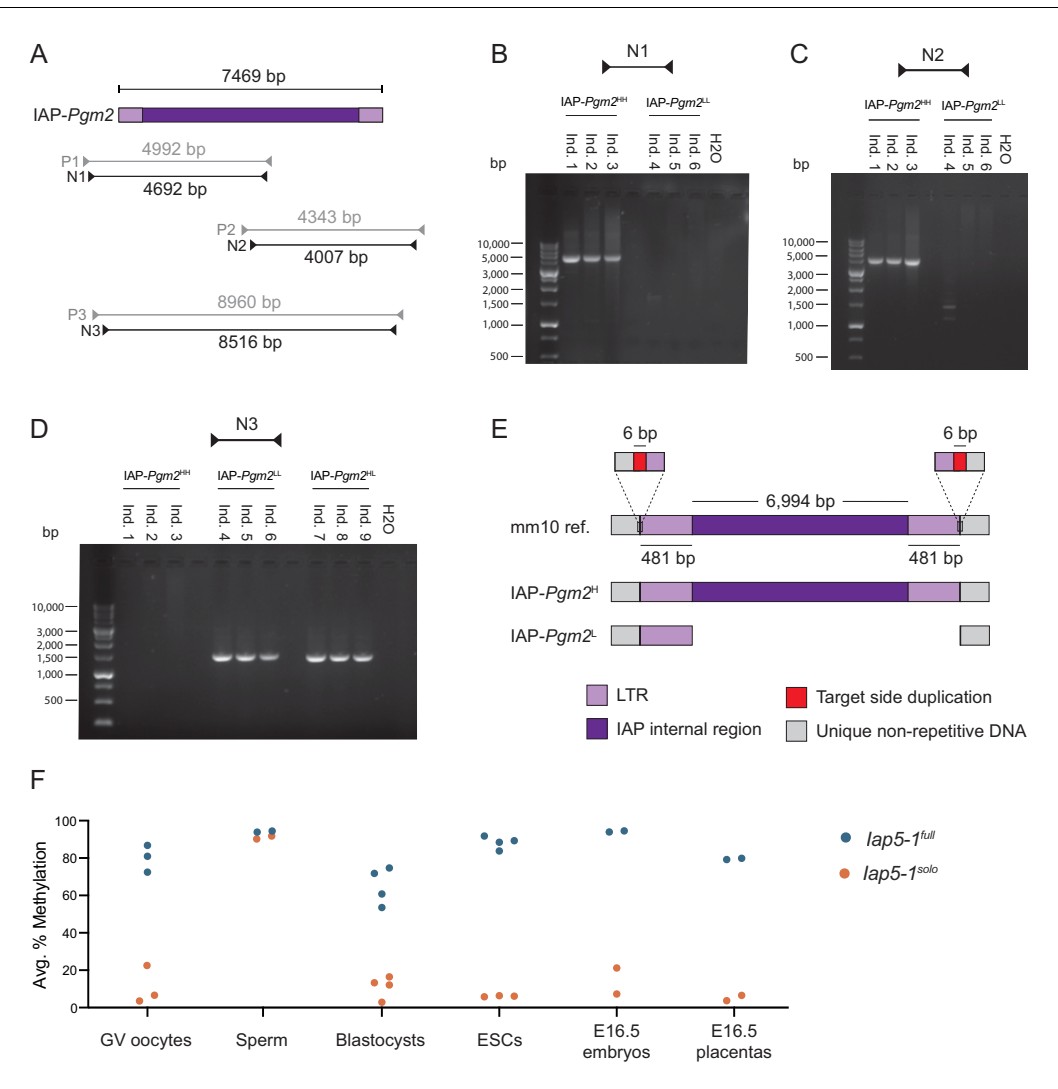

**Figure 2.** The intracisternal A-particle (IAP)-*Pgm2*L allele is a solo long terminal repeat (LTR) formed via inter-LTR homologous recombination. (**A**) Map of PCR primer pairs P1–P3 and nested PCR primer pairs N1–N3 (not drawn to scale). (**B and C**) Agarose gel electrophoresis of PCR products amplified from three IAP-*Pgm2*HH and three IAP-*Pgm2*LL DNA samples using primer pairs N1 and N2. (**D**) As in panels **B** and **C**, but using primer pair N3 and including three IAP-*Pgm2*HL DNA samples. (**E**) Schematic representation of the alignment of the IAP-*Pgm2* mm10 reference sequence and the assembled IAP-*Pgm2*H and IAP-*Pgm2*L sequences following Sanger sequencing (not drawn to scale). The IAP-*Pgm2*L solo LTR could have equivalently been shown aligned to the 3′ LTR because the 5′ and 3′ LTRs have identical sequences. The base-resolution sequence of the new IAP-*Pgm2*L allele, renamed *Iap5-1*solo, is available on GenBank (accession number: MW308129). (**F**) Both *Iap5-1* variants are methylated in the male germline, with differential methylation re-established in early embryonic development. DNA methylation levels at the *Iap5-1*full and *Iap5-1*solo alleles were quantified in oocytes, sperm, blastocysts, ESC lines, and E16.5 embryonic tail and placenta samples. Respective data points represent the following: 100 pooled oocytes, sperm collected from one male, pooled littermate blastocysts, one ESC line, and individual E16.5 embryos and placentas. All data points represent average DNA methylation across the four most distal CpGs of the 5′ end of *Iap5-1*.

The online version of this article includes the following source data and figure supplement(s) for figure 2:

**Source data 1.** Numerical data represented in panel F.

**Figure supplement 1.** Possible mechanisms of homologous recombination between intracisternal A-particle (IAP) long terminal repeats (LTRs) leading to the formation of a solo LTR at the *Iap5-1* locus.

**Figure supplement 2.** Differential DNA methylation between the *Iap5-1*full and *Iap5-1*solo variants extends over the entire long terminal repeat(s) (LTR[s]).

To determine the location of the 7 kb deletion, we purified and sequenced the PCR products and aligned the assembled IAP-*Pgm2*[H] and IAP-*Pgm2*[L] sequences. The alignment revealed that the IAP-*Pgm2*[H] allele is an identical match to the full-length IAP-*Pgm2* sequence from the mm10 reference, while the IAP-*Pgm2*[L] allele is a solo LTR (*Figure 2E*). The IAP-*Pgm2*[L] solo LTR and the IAP-*Pgm2*[H] full-length IAP are both flanked by the same target site duplications (TSDs) and the IAP-*Pgm2*[L] solo LTR exhibits 100% sequence identity to both the 5′ and 3′ LTRs of the full-length IAP-*Pgm2*[H] (*Figure 2E*). Therefore, a recent inter-LTR homologous recombination event in the inbred B6 mouse strain gave rise to the solo LTR variant. In accordance with nomenclature guidelines from the International Committee on Standardized Genetic Nomenclature for Mice (ICSGNM), IAP-*Pgm2*[H] and IAP-*Pgm2*[L] are hereafter referred to as *Iap5-1*^full^ and *Iap5-1*^solo^, respectively.

Inter-LTR recombination events occur when identical or near-identical LTR sequences engage in ectopic homologous recombination, resulting in the formation of solo LTRs (*Jern and Coffin, 2008*). These can occur both intra- and inter-chromosomally (*Figure 2—figure supplement 1*). Although we identified both *Iap5-1* allelic variants in our own B6 colony, we also detected them in B6 samples sent to us from mainland Europe and North America (data not shown), so the recombination event most likely occurred at a B6-distributing facility. This type of proviral excision event is common: approximately half of the IAPs in the mouse genome are solo LTRs (*Nellåker et al., 2012*). However, the vast majority of the ~5,000 solo LTRs in the B6 genome are highly methylated (*Shimosuga et al., 2017*), making *Iap5-1*^solo^ unique from a regulatory perspective and raising questions regarding the functional consequences of a solo LTR left unmodified. Using clonal bisulphite sequencing, we confirmed that the entirety of the *Iap5-1*^solo^ solo LTR is unmethylated (bar the occasional methylated CpGs at the 5′ end, consistent with the pyrosequencing data) while all CpGs in both the 5′ and 3′ LTRs of *Iap5-1*^full^ are highly methylated (*Figure 2—figure supplement 2*).

## Developmental dynamics of *Iap5-1* methylation states

The experiments described thus far have focused on adult somatic *Iap5-1* DNA methylation within and across generations. Given the dynamic nature of DNA methylation during mammalian development and considering previous reports on the resistance of IAPs to genome-wide methylation erasure (*Lane et al., 2003*; *Seisenberger et al., 2012*), we sought to examine and compare DNA methylation levels of the *Iap5-1*^full^ and *Iap5-1*^solo^ variants in the germline and during early embryonic development.

Germinal vesicle (GV) oocytes and mature sperm were collected from *Iap5-1*^full^ and *Iap5-1*^solo^ adult B6 females and males. Oocytes collected from *Iap5-1*^full^ and *Iap5-1*^solo^ females were highly and lowly methylated at the *Iap5-1* locus, respectively (*Figure 2F*). Thus, oocyte *Iap5-1* methylation levels are reflective of somatic *Iap5-1* methylation levels. In contrast, both variants were hypermethylated in sperm (*Figure 2F*), indicating that both alleles are targeted for repression during spermatogenesis by mechanism(s) that are distinct from those operating on *Iap5-1* in the soma. Together, these experiments indicate that the maternal and paternal *Iap5-1*^solo^ alleles are differentially methylated upon fertilisation in the early zygote.

We examined the behaviour of *Iap5-1* methylation levels during early development in blastocysts (embryonic day 3.5, E3.5) collected from the uteri of *Iap5-1*^full^ and *Iap5-1*^solo^ B6 females bred to *Iap5-1*^full^ and *Iap5-1*^solo^ B6 males, respectively. Blastocysts generated from *Iap5-1*^solo^ parents exhibited low methylation levels at *Iap5-1*, suggesting that the paternally inherited hypermethylated *Iap5-1*^solo^ allele becomes demethylated shortly after fertilisation (*Figure 2F*). By comparison, blastocysts generated from *Iap5-1*^full^ parents exhibited methylation levels around 70% (*Figure 2F*). It is unclear whether *Iap5-1*^full^ is demethylated after fertilisation and rapidly methylated again by the blastocyst stage, or whether *Iap5-1*^full^ is generally resistant to epigenetic reprogramming during preimplantation development. Nonetheless, these results demonstrate that methylation patterns at *Iap5-1*^full^ and *Iap5-1*^solo^ are specified prior to implantation.

Despite the marked distinction in methylation states between *Iap5-1*^full^ and *Iap5-1*^solo^ blastocysts, *Iap5-1*^full^ methylation levels are lower at the blastocyst stage than in adult somatic tissue. It is possible that this incomplete methylation is symptomatic of lower methylation levels in the developing trophectoderm which counteract the higher methylation levels in the inner cell mass (ICM). In support of this, DNA methylation levels in embryonic stem (ES) cell lines derived from the ICM of *Iap5-1*^full^ and *Iap5-1*^solo^ blastocysts closely matched those observed in adult somatic tissues, with ES cells derived from *Iap5-1*^full^ blastocysts nearing 100% methylation (*Figure 2F*). In addition, we found that

*Iap5-1^full* methylation levels in E16.5 placentas were lower than those in E16.5 embryonic tissue (*Figure 2F*), consistent with reduced global DNA methylation in the placenta (*Ehrlich et al., 1982*; *Schroeder et al., 2015*). These results provide evidence for differential methylation between ICM- and trophectoderm-derived lineages at the *Iap5-1* locus. Of note, even though placental *Iap5-1* methylation levels are less methylated compared to their embryonic counterparts, the two variants retain a pronounced difference in DNA methylation levels in this tissue.

## Loss of H3K9me3 occupancy at *Iap5-1^solo^*

To determine whether loss of DNA methylation at *Iap5-1^solo^* is associated with changes in histone modifications, we quantified H3K9me3 enrichment in *Iap5-1^full^* and *Iap5-1^solo^* adult liver samples via ChIP-qPCR. The retrotransposons IAP-*Asxl3* and SINE-*Rbak*, used as positive controls, exhibited equivalent H3K9me3 enrichment between *Iap5-1^full^* and *Iap5-1^solo^* individuals (*Figure 3A*). The *Gapdh* promoter was used as a negative control. Because the sequence of *Iap5-1^solo^* is identical to that of the 5′ and 3′ LTRs of *Iap5-1^full^*, the same primers were used to probe H3K9me3 enrichment at the borders of both variants. H3K9me3 enrichment at the 5′ and the 3′ borders was significantly decreased in *Iap5-1^solo^* samples compared to *Iap5-1^full^* samples (*Figure 3A*). *Iap5-1^full^* showed comparable H3K9me3 levels to those observed at the positive controls (*Figure 3A*). We suggest that the slightly higher H3K9me3 enrichment at the 3′ end compared to the 5′ end in *Iap5-1^solo^* samples is due to the presence of an ERV element immediately downstream of *Iap5-1*.

Heterochromatin formation at mammalian TEs occurs in early development following their sequence-specific recognition by KZFPs. KZFPs recruit the scaffold protein KAP1, which in turn recruits the H3K9 methyltransferase SETDB1 as well as de novo DNA methyltransferases (*Ecco et al., 2017*). We reasoned that *Iap5-1^full^* may be targeted for repression in the early embryo by KZFP(s) whose binding sites are located in the internal proviral portion of the IAP which is no longer present in the *Iap5-1^solo^* variant. To explore this hypothesis, we analysed previously published ChIP-seq (chromatin immunoprecipitation followed by high-throughput sequencing) binding profiles of more than 60 murine KZFPs (*Wolf et al., 2020*) and identified two *Iap5-1^full^*-binding candidates, Gm14419 and Gm8898 (*Figure 3B*). The ChIP-seq profiles for Gm14419 and Gm8898 in B6 ES cells displayed peaks immediately downstream and 1.5 kb downstream of the 5′ LTR, respectively (*Figure 3B*). We observed KAP1 occupancy at the Gm14419 binding site, rendering Gm14419 a particularly promising candidate for future mechanistic research (*Figure 3B*). The Gm14419 peak overlaps with the primer binding site (PBS) just downstream of the 5′ LTR (*Figure 3B*), a retroviral sequence often bound by KZFPs to effectuate repression (*Wolf and Goff, 2007*). Furthermore, it is likely that additional as yet unidentified KZFP(s) bind to *Iap5-1^full^*, as evidenced by a second KAP1 peak in the *Iap5-1^full^* internal region which does not overlap with the binding site of any of the KZFPs identified in our analysis (*Figure 3B*). This is consistent with the redundant nature of KZFP-mediated ERV repression (*Imbeault et al., 2017*; *Wolf et al., 2020*). In addition, we detected ZFP429 binding peaks at the 5′ and 3′ LTRs of *Iap5-1^full^* (*Figure 3B*). Given the shared sequence between the *Iap5-1^full^* LTRs and *Iap5-1^solo^*, this observation suggests that ZFP429 does not recruit heterochromatin factors as effectively as other KZFPs. This is in line with our previous finding that ZFP429 is enriched at VM-IAPs compared to fully methylated IAPs of the same subclass (*Bertozzi et al., 2020*).

## Genetic background influences *Iap5-1^solo^* methylation levels

Previous studies have shown that in some cases IAP methylation is modulated by genetic background (*Bertozzi et al., 2020*; *Elmer and Ferguson-Smith, 2020*; *Rakyan et al., 2003*; *Wolff, 1971*). To assess whether this is the case for *Iap5-1* methylation, we carried out reciprocal crosses between B6 and wild-derived CAST/EiJ (CAST) mice (BC, B6 female × CAST male; CB, CAST female × B6 male). F1 hybrid offspring carry a single copy of *Iap5-1* inherited from their B6 parent because the *Iap5-1* insertion is absent from the CAST genome (*Figure 4A*). In line with our breeding experiments in pure B6 mice, hemizygous offspring born to a *Iap5-1^full^* B6 parent were highly methylated and those born to a *Iap5-1^solo^* B6 parent were lowly methylated (*Figure 4B*). However, although *Iap5-1^solo^* methylation levels remained low in BC and CB F1 hybrids, they were significantly higher compared to those in pure B6 individuals (*Figure 4C*), suggesting that CAST-derived modifier(s) act on *Iap5-1^solo^* in trans. In addition, CB offspring displayed higher and more variable *Iap5-1^solo^*

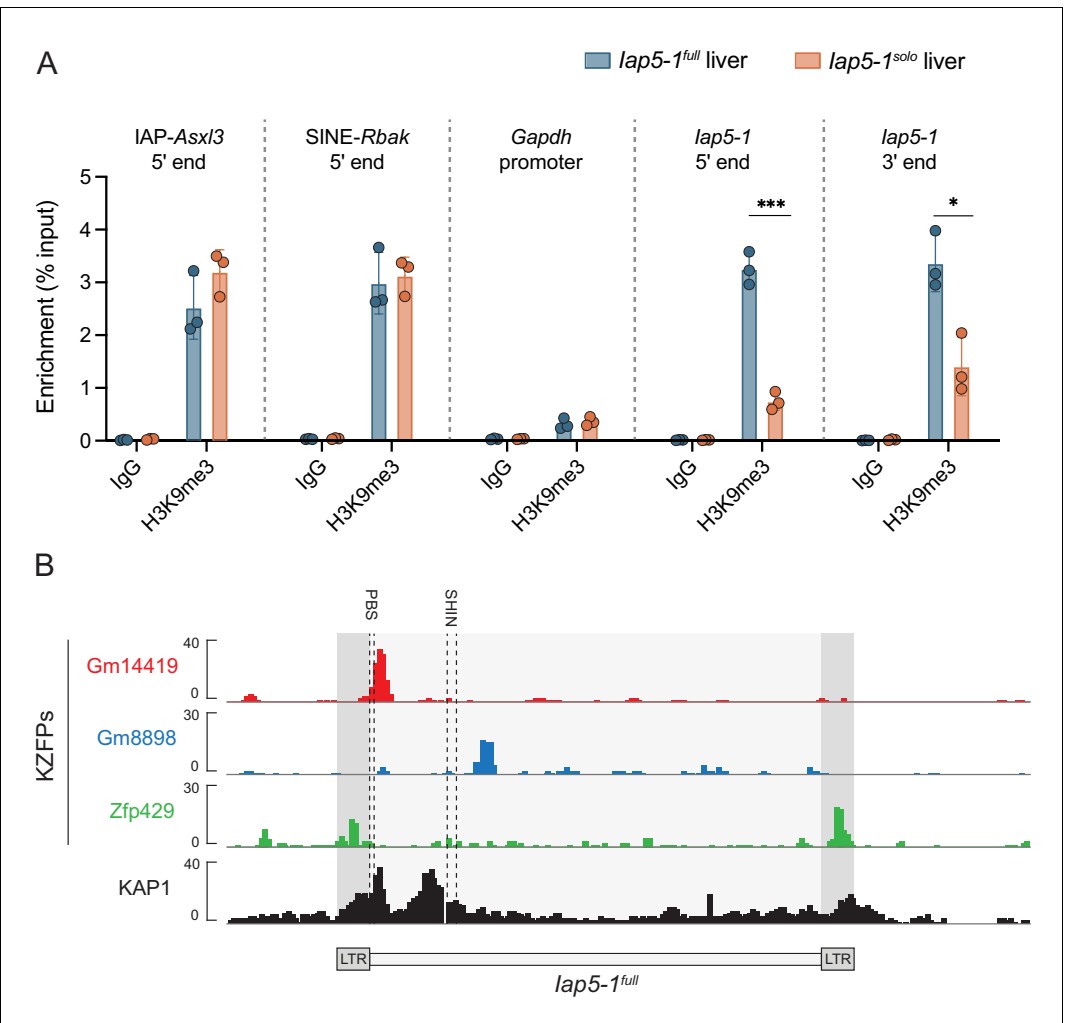

**Figure 3.** Lack of DNA methylation at *Iap5-1^solo^* is accompanied by a loss of H3K9me3 marks. (**A**) H3K9me3 ChIP-qPCR on *Iap5-1^full^* and *Iap5-1^solo^* adult male livers. IAP-*Asxl3* and SINE-*Rbak* are positive control loci, the *Gapdh* promoter is a negative control locus, and the Rabbit IgG antibody serves as a negative isotype control in the two B6 populations. H3K9me3 enrichment was calculated using the per cent input method and compared between genotypes using unpaired t-tests (*p<0.05; ***p<0.0005). Error bars represent standard deviations of biological replicates. (**B**) *Iap5-1^full^* is bound by multiple KZFPs. Publicly available ChIP-seq data sets indicate that KZFPs Gm14419 and Gm8898 are capable of binding the internal region of *Iap5-1^full^* and that KZFP Zfp429 is capable of binding the long terminal repeat (LTR) sequence shared by both *Iap5-1^full^* and *Iap5-1^solo^*. The primer binding site (PBS) and the short heterochromatin inducing sequence (SHIN), both frequently bound by KZFPs, are shown with dashed lines. ChIP-seq data sets were downloaded from the GEO database (accession numbers: Gm14419, GSM3173720; Gm8898, GSM3173728; Zfp429, GSM3173732; KAP1, sum of GSM3173661, GSM3173662, and GSM3173663).

The online version of this article includes the following source data for figure 3:

**Source data 1.** Numerical data represented in panel A.

methylation levels compared to BC offspring, indicative of a genetic background-specific maternal effect similar to those recently reported at VM-IAPs (*Bertozzi et al., 2020*).

We further interrogated the strain-specific maternal effect by backcrossing *Iap5-1^solo^* CB hybrid males to pure CAST females. This resulted in a cumulative effect, whereby N1 *Iap5-1^solo^* mice showed even higher and more variable methylation levels compared to F1 *Iap5-1^solo^* mice (*Figure 4C*). Therefore, the CAST content of the inherited paternal genome and passage through a CAST egg may compound each other. The subsequent N2 backcrossed generation did not cause a further increase in methylation, and backcrossing N5 males to B6 females resulted in a reversion to

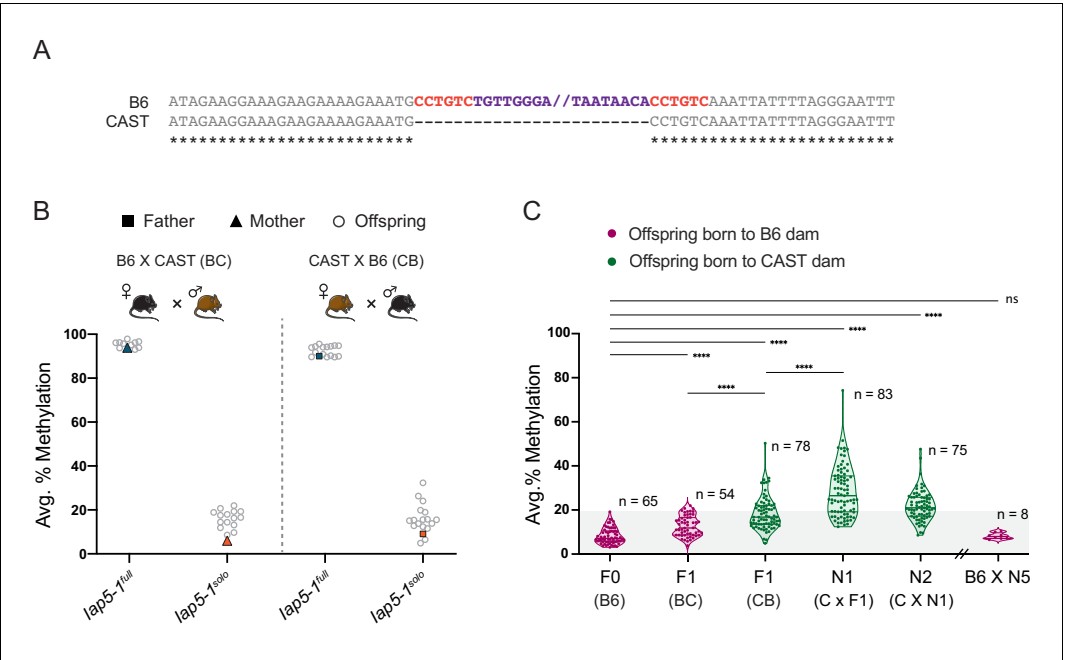

**Figure 4.** Genetic background influences *Iap5-1^solo* methylation levels. (**A**) B6 and CAST sequence alignment of the *Iap5-1* insertion site reveals that the *Iap5-1* element (purple) and its associated target site duplication (TSD) sequence (red) are not present in the CAST genome. Asterisks indicate nucleotide conservation. The internal portion of the B6 sequence was omitted; nothing was omitted from the CAST sequence. (**B**) Reciprocal F1 hybrids uncover a variant-specific genetic background effect. *Iap5-1^full^* and *Iap5-1^solo^* B6 females were crossed to CAST males (BC, left); *Iap5-1^full^* and *Iap5-1^solo^* B6 males were crossed to CAST females (CB, right). DNA methylation levels were quantified in ear samples collected from hemizygous offspring. Each data point represents average DNA methylation across the four distal CpGs of the 5′ end of *Iap5-1* for one individual. (**C**) The genetic background-specific maternal effect at *Iap5-1^solo^* is strengthened upon backcrossing to a CAST female and lost following backcross to a B6 female. The N1 generation was produced by crossing F1 CB males to CAST females, and the N2 generation was generated by crossing N1 males harbouring the *Iap5-1^solo^* allele to CAST females. After five generations of backcrossing to CAST, N5 males were crossed to B6 females. Offspring that did not inherit the *Iap5-1^solo^* allele were not included in the analysis. Thick and thin lines in the violin plots designate the median and distribution quartiles, respectively. Sample sizes are shown on the graph. Grey shaded area represents the range of methylation levels observed in B6 individuals. Statistics: Welch's ANOVA test followed by Games-Howell's post hoc multiple comparison test (****$p<0.0001$; ns: not significant).

The online version of this article includes the following source data and figure supplement(s) for figure 4:

**Figure supplement 1.** DNA methylation at *Iap5-1^solo^* is unresponsive to maternal dietary methyl supplementation.

**Figure supplement 1—source data 1.** Numerical data represented in panels B, C, and *Figure 4—figure supplement 1*.

the original B6 methylation state (*Figure 4C*). These results are reminiscent of strain-specific behaviours of transgene methylation reported decades ago and support a role for strain-specific oocyte factors (possibly polymorphic KZFPs) in driving these genetic–epigenetic interactions (*Allen et al., 1990*; *Bertozzi et al., 2020*; *Kearns et al., 2000*).

## *Iap5-1^solo^* methylation is not affected by maternal dietary methyl supplementation

The IAPs located at the *A^vy^* and *Axin^Fu^* VM-IAPs exhibit increased DNA methylation following gestational methyl supplementation, resulting in shifts in the associated coat colour and tail morphology phenotypes, respectively (*Waterland et al., 2006*; *Waterland and Jirtle, 2003*). However, it remains unclear whether susceptibility to this environmental exposure is conferred by *variable* or by *incomplete* DNA methylation at these epialleles (or neither). The unmethylated *Iap5-1^solo^* variant provides an opportunity to test the latter.

To examine whether *Iap5-1^solo* is responsive to methyl supplementation, *Iap5-1^solo* females were put on a methyl-supplemented diet 2 weeks prior to mating and were kept on the same diet throughout pregnancy and lactation. Control females were fed standard chow throughout and all pups in the experiment were weaned onto standard chow. DNA methylation levels at the *Iap5-1^solo* LTR were quantified in 8-week-old offspring liver samples. Unlike *A^vy* and *Axin^Fu* loci, *Iap5-1^solo* remained unmethylated in the methyl-supplemented offspring (*Figure 4—figure supplement 1*), indicating that complete lack of methylation at IAPs does not go hand-in-hand with susceptibility to dietary methyl supplementation.

## *Iap5-1* variants influence adjacent gene expression in a tissue-specific manner

IAP insertions can influence neighbouring gene expression. Intergenic IAPs can induce the formation of chimeric transcripts initiated at the promoter in the IAP LTR and may also act as enhancers (*Gagnier et al., 2019*). These effects are sometimes dependent on the epigenetic properties of the IAP, as illustrated by the *A^vy* and *Axin^Fu* IAPs which modulate *Agouti* or *Axin* expression in a methylation-dependent manner.

We asked whether allelic variation at the *Iap5-1* locus influences the expression of neighbouring genes by quantifying expression of the four closest genes (*Figure 5A*) in liver, cortex, thymus, and placental tissues collected from *Iap5-1^full* and *Iap5-1^solo* individuals. Protein-coding genes *Pgm2*, *TBC1 domain family, member 1* (*Tbc1d1*), and *RELT-like protein 1* (*Rell1*) were expressed in all tissues examined; long non-coding RNA (lncRNA) *5830416I19Rik* transcripts were only detected in the thymus. *Pgm2* and *Rell1* expression levels in the thymus and placenta were significantly higher in *Iap5-1^solo* than in *Iap5-1^full* individuals, indicating that in these tissues the unmethylated solo LTR is associated with increased expression (*Figure 5B*). *Tbc1d1*, the furthest in distance from *Iap5-1*, did not display significant differences in expression in any of the tested tissues, suggesting that proximity to the *Iap5-1* locus is predictive of its transcriptional effect (*Figure 5B*). *5830416I19Rik* expression in the thymus was barely detected in *Iap5-1^full* samples, showing a highly significant increase in *Iap5-1^solo* samples (*Figure 5B*). *Rell1* was the only gene to show a significant difference in expression in liver tissue and the directionality of the effect was inversed, with lower expression levels observed in *Iap5-1^solo* individuals (*Figure 5B*). No significant differences in expression were observed for any of the genes in cortex samples (*Figure 5B*). In thymus and cortex, however, we detected transcription of the unique non-coding regions bordering *Iap5-1^solo*, which was not observed for the equivalent regions bordering *Iap5-1^full* (*Figure 5—figure supplement 1*). Together, these data show that the *Iap5-1* polymorphism is associated with tissue-specific altered adjacent gene expression.

## The formation of the *Iap5-1^solo* variant is associated with metabolic phenotypes

Little is known about the biological functions of *Rell1* and *5830416I19Rik*, but *Pgm2* is better characterised. *Pgm2* is the closest coding gene to *Iap5-1*, lying 55 kb downstream of the 3′ LTR. PGM2 catalyses the interconversion between glucose 1-phosphate and glucose 6-phosphate and has a secondary role to that of the predominant PGM isozyme, PGM1 (*Geer et al., 2010*). PGM1 deficiency in humans is associated with glycogen storage disease and a congenital disorder of glycosylation (*Beamer, 2015*). PGM2 has been associated with metabolic disease in GWAS studies (*Timmons et al., 2018*).

Considering the established role of *Pgm2* in metabolic pathways, we investigated potential metabolic effects resulting from the formation of the *Iap5-1^solo* allele. We screened plasma samples collected from *Iap5-1^full* and *Iap5-1^solo* adult males for a range of metabolic biomarkers. While most assays revealed no difference between the two variants (*Figure 5—figure supplement 2A*), we found that fasting plasma glucose and triglyceride concentrations were significantly lower in *Iap5-1^solo* mice (*Figure 5C*). Glucose tolerance tests did not show additional metabolic disparities between the two variants (*Figure 5—figure supplement 2B*). These findings suggest that the emergence of a derepressed solo LTR near a gene involved in glucose metabolism has phenotypic implications, providing a basis for further functional characterisation of this model.

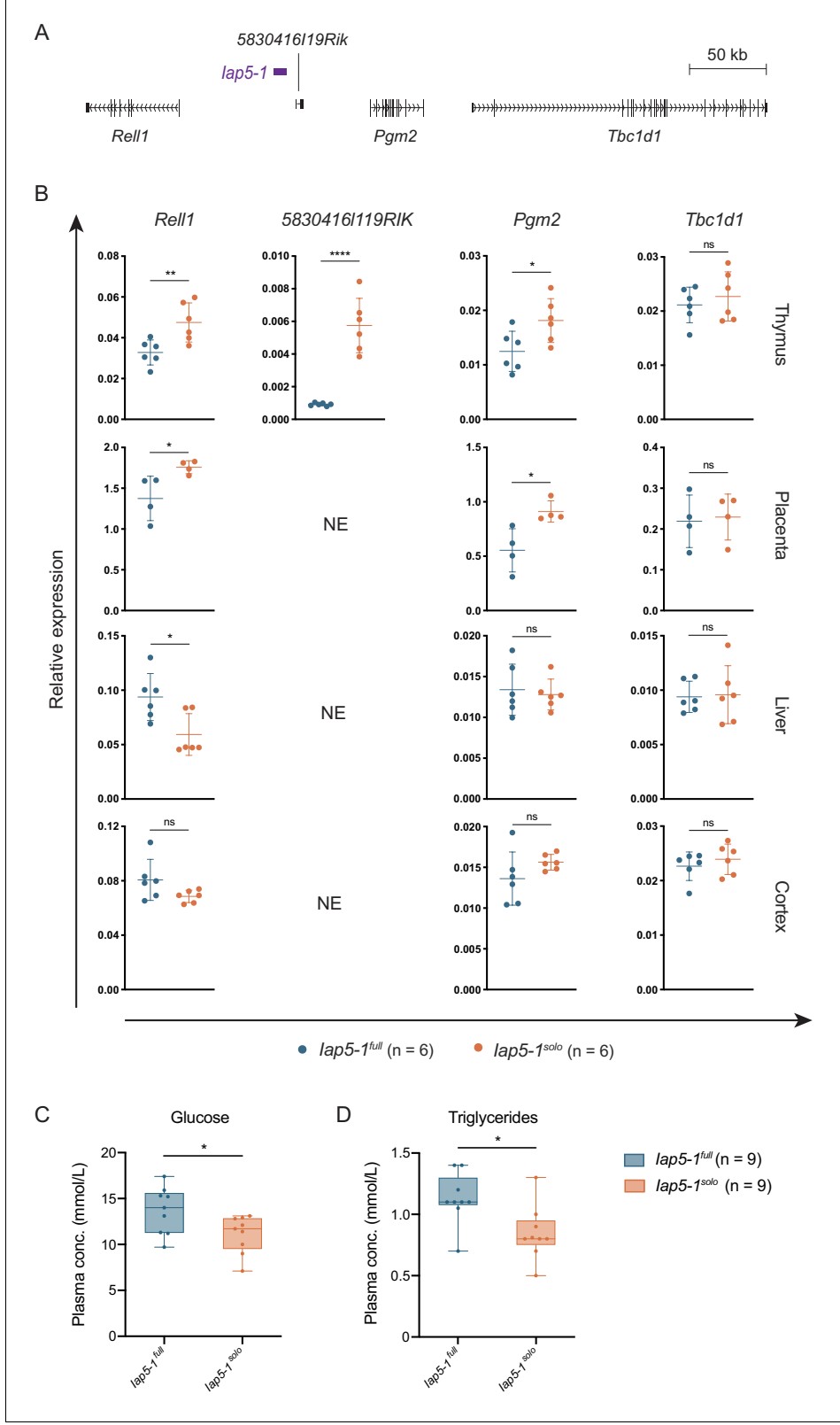

**Figure 5.** Functional consequences of the genetically induced epiallele at the *Iap5-1* locus. (**A**) Scaled map of the genes surrounding *Iap5-1*. Gene transcripts were extracted from the UCSC Genome Browser (*Haeussler et al., 2019*). (**B**) The *Iap5-1* polymorphism influences neighbouring gene expression in a tissue-specific manner. Expression of *Rell1* (exon 7), *5830416l19Rik* (exon 3), *Pgm2* (exon 9), and *Tbc1d1* (exon 3) was quantified in *Iap5-*

*Figure 5 continued*

1*full* and *Iap5-1solo* thymus, placenta, liver, and brain tissues via RT-qPCR (NE: not expressed). Relative expression was normalised to *Hprt1* and *β-actin* expression and calculated using the ΔCt method. Means and standard deviations of biological replicates are shown in the graphs. (C and D) Plasma glucose (C) and triglyceride (D) concentrations in *Iap5-1full* and *Iap5-1solo* adult males. Box plots show the distribution quartiles and median. Statistics: unpaired t-tests (*p<0.05; **p<0.005; ****p<0.0001; ns: not significant).

The online version of this article includes the following source data and figure supplement(s) for figure 5:

**Source data 1.** Numerical gene expression data represented in *Figure 5B* and *Figure 5—figure supplement 1B*.
**Source data 2.** Numerical data for metabolic experiments shown in *Figure 5C and D* and *Figure 5—figure supplement 2*.
**Figure supplement 1.** Bordering genomic regions are transcribed in *Iap5-1solo* tissues.
**Figure supplement 2.** Metabolic phenotyping of *Iap5-1full* and *Iap5-1solo* individuals.

## Discussion

Historically, inter-LTR recombination events have been identified due to phenotypic reversions of ERV-induced mutations, whereby the phenotypic effect of the insertion is reversed following proviral excision and solo LTR formation (*Bultman et al., 1994*; *Seperack et al., 1988*; *Stoye et al., 1988*). In this study, we identified one such chromosomal event based on its epigenetic rather than phenotypic outcome, revealing *Iap5-1* as a genetically induced epiallele in the ostensibly isogenic B6 mouse strain. We demonstrated that the *Iap5-1full* and *Iap5-1solo* variants display distinct DNA and H3K9 methylation profiles which are associated with differential adjacent gene expression and altered metabolism, establishing the *Iap5-1* locus as a valuable endogenous system to study the mechanisms, evolution, and functional implications of TE repression in both the germline and soma.

The unmethylated status of *Iap5-1solo* sets it apart from other solo LTRs in the B6 genome, the vast majority of which (>93%) are highly methylated (*Shimosuga et al., 2017*). Shimosuga and colleagues quantified DNA methylation at more than 8000 B6 IAP LTRs and only 14 exhibited less than 20% methylation in tail samples. Ten of these were solo LTRs, mostly of the same subtype as the *Iap5-1* LTR (IAPLTR2_mm). While the authors classified the *Iap5-1* LTR as a hypomethylated 3' LTR of a full-length IAP, in all likelihood they unknowingly detected the hypomethylated *Iap5-1solo* allele in their particular samples. It is conceivable that some of the other documented hypomethylated LTRs of full-length IAPs are in fact solo LTRs formed from recent recombination events. In comparing tail and sperm DNA methylation, the same study showed that individual hypomethylated IAPs in either tail or sperm were non-overlapping. This is consistent with the hypermethylation of *Iap5-1solo* that we observed in sperm and highlights the divergence of TE silencing mechanisms between the soma and the germline.

Ancestral variants of spontaneous mutations in inbred mouse strains are easily lost from the population. The fortuitous timing of our discovery and the resultant availability of both *Iap5-1full* and *Iap5-1solo* mice allowed us to quantitatively explore the functional repercussions of a derepressed solo LTR. We showed that the *Iap5-1solo* variant is associated with increased expression of adjacent genes in thymic and placental tissue. This may reflect the use of newly accessible regulatory sequences in the solo LTR or, alternatively, a secondary effect stemming from heterochromatin loss at a neighbouring region. The transcriptional consequences were tissue-specific: *Rell1* expression was lower in *Iap5-1solo* compared to *Iap5-1full* livers, and no gene expression differences were observed in cortex samples. The lncRNA *5830416I19Rik*, identified from the sequencing of adult male thymus and aorta cDNA libraries (*Kawai et al., 2001*), was only expressed in thymus samples collected from *Iap5-1solo* individuals; it was nearly undetectable in *Iap5-1full* thymic samples and not expressed at all in any of the other tested tissues. We speculate that the formation of *Iap5-1solo* gave rise to *5830416I19Rik* as a novel tissue-specific (perhaps immune-specific) lncRNA. Indeed, IAP transcripts are rarely detected in somatic tissues outside of tumours and early embryogenesis, but thymic and activated splenic cells are notable exceptions (*Kuff and Lueders, 1988*). In addition, young ERVs have recently been implicated in shaping the evolution of transcriptional networks underlying innate immunity (*Chuong et al., 2016*; *Tie et al., 2018*; *Ye et al., 2020*), and our group recently identified individual IAP elements that exhibit inter-individual methylation variability exclusively in B cells (*Elmer et al., 2021*).

Along with the transcriptional differences observed at the metabolic gene *Pgm2*, we found that *Iap5-1^{solo}* individuals exhibit lower plasma glucose and triglyceride concentrations compared to *Iap5-1^{full}* individuals. While the effect sizes were small, we note that these measurements were taken in an unaltered environment. Administration of a long-term metabolic challenge such as a high-fat diet and subsequent metabolic phenotyping will aid in further elucidating the biological significance of these results. Nevertheless, our work adds to the body of work suggesting that TE insertions and their epigenetic modification can have quantifiable consequences on host metabolic pathways (*Du et al., 2016*; *Kuehnen et al., 2012*; *Scherneck et al., 2009*; *Yen et al., 1994*).

We have shown that *Iap5-1^{solo}* is susceptible to genetic background effects, whereby passage through a CAST egg promotes methylation of the B6-derived solo LTR. In a recent study we reported the same effect at a number of VM-IAPs and showed that KZFP diversification is associated with strain-specific IAP methylation (*Bertozzi et al., 2020*). This combined with the identification of candidate KZFPs capable of binding the *Iap5-1* variants suggests that the differential recruitment of heterochromatin factors by KZFPs is involved in both driving the epigenetic differences between *Iap5-1^{full}* and *Iap5-1^{solo}*, and for the strain-specific methylation of *Iap5-1^{solo}*. In fact, by definition, the wide range of methylation levels observed at the *Iap5-1^{solo}* allele across genetically identical CB individuals renders this locus a bona fide VM-IAP (or *metastable epiallele*) in this hybrid context (*Bertozzi and Ferguson-Smith, 2020*; *Rakyan et al., 2002*). Therefore, we expect the new *Iap5-1^{solo}* variant to be a useful tool for the study of epigenetic stochasticity in the absence of genetic variation, a phenomenon for which the underlying mechanisms remain poorly understood.

The identification of *Iap5-1^{solo}* demonstrates that cryptic genetic diversity in an inbred mouse population can have functional repercussions with important implications for experimental outcomes. It serves as a cautionary tale for researchers working with inbred mouse colonies, particularly in the field of epigenetics where ruling out genetic effects is often paramount. We are reminded that the current mouse reference genome harbours large gaps and inaccuracies, largely due to mapping difficulties associated with the repeat genome. Inter-LTR recombination events are surely an underappreciated source of genetic variation considering that a lack of uniquely mapped reads in internal portions of TEs is more likely to be attributed to technical rather than biological limitations. A recent study in humans developed a computational pipeline to capture dimorphic human ERVs (HERVs) resulting from inter-LTR recombination events and detected dozens of previously unidentified candidates (*Thomas et al., 2018*). The advent of such analytical tools as well as routine long-read sequencing of whole genomes will be highly beneficial in addressing these issues.

In summary, we have identified and characterised a recent spontaneous inter-LTR recombination event at the *Iap5-1* locus, introducing a genetic variant in the commonly investigated and supposedly inbred B6 mouse strain. The ancestral variant, *Iap5-1^{full}*, is a full-length IAP repressed by DNA methylation and H3K9 trimethylation, as is typical for this class of evolutionary young ERV. The recently formed solo LTR variant, *Iap5-1^{solo}*, lacks these silencing marks and is associated with metabolic changes and differential neighbouring gene expression. Our study lays the foundation for further comparative studies between the *Iap5-1^{full}* and *Iap5-1^{solo}* variants aimed at better understanding the epigenetic and functional consequences of TE structural variation and evolution.

# Materials and methods

## Key resources table

| Reagent type (species) or resource | Designation | Source or reference | Identifiers | Additional information |
|---|---|---|---|---|
| Strain, strain background (*Mus musculus*) | C57BL/6J | The Jackson Laboratory via Charles River | 000664 (JAX) 632 (Charles River) | |
| Strain, strain background (*Mus musculus*) | CAST/EiJ | MRC Harwell Institute | FESA:773 | |

*Continued on next page*

*Continued*

| Reagent type (species) or resource | Designation | Source or reference | Identifiers | Additional information |
|---|---|---|---|---|
| Cell line (*Mus musculus*) | C57BL6/J ES cell lines | This paper | | See Materials and methods, Generation of ESC lines. |
| Antibody | Rabbit polyclonal anti-H3K9me3 | Active Motif | RRID:AB_2532132 | ChIP (2.5 µl/30 mg liver tissue) |

## Mice

Mouse work was carried out in accordance with the Animals (Scientific Procedures) Act 1986 Amendment Regulations 2012 following ethical review by the University of Cambridge Animal Welfare and Ethical Review Body (Home Office project license # PC213320E). C57BL/6J (RRID:IMSR_JAX:000664) and CAST/EiJ (RRID:IMSR_JAX:000928) mice were obtained from Charles River and the MRC Harwell Institute, respectively, and maintained under a 12 hr light–dark cycle in temperature- and humidity-controlled conditions. Mice were fed a standard chow diet (RM3(E); Special Diet Services) ad libitum unless otherwise noted. For pure and hybrid breeding experiments, mice were mated at 8–12 weeks of age and 10 day old pups were ear notched for DNA methylation quantification. Male and female offspring from multiple litters per breeding pair were included in the analyses.

## Dietary methyl supplementation

Adult *Iap5-1^{solo}* females were randomly placed on control diet (RM3(E); Special Diet Services) or methyl supplemented diet (RM3(E) supplemented with 15 g Choline, 15 g Betaine, 15 mg Folic acid, 1.5 mg Vitamin B12, 7.5 g L-methionine, 150 mg Zinc; Special Diet Services) 2 weeks prior to mating with *Iap5-1^{solo}* males and kept on their respective diets throughout pregnancy and lactation. Eight dams were used for each dietary group. The methyl supplemented diet recipe matches the 3SZM diet in *Wolff et al., 1998*. All offspring were weaned onto the control diet and culled at 8 weeks of age. DNA methylation was quantified at the distal CpGs of the 5′ end of *Iap5-1^{solo}* in all offspring livers (control diet: 38 born from eight litters; methyl supplemented diet: 41 born from eight litters). Statistics were carried out on litter averages because the dietary intervention was done on the dams, not the offspring.

## Metabolic biomarker assays

Blood was sampled from 4-month-old *Iap5-1^{full}* and *Iap5-1^{solo}* males via cardiac puncture following isoflurane-induced deep terminal anaesthesia. Blood samples were placed in EDTA-coated tubes and centrifuged at 1500 × g for 15 min at 4°C. Plasma was collected from the separated upper phase and stored at −80°C before sending to the Core biochemical assay laboratory (CBAL) at the Cambridge University Hospitals for analysis. Assays performed on the plasma samples included adiponectin (ng/ml), albumin (g/l), ALT (U/l), cholesterol (mmol/l), corticosterone (ng/ml), glucagon (pg/ml), glucose (mmol/l), HDL (mmol/l), insulin (µg/l), LDL (mmol/l), leptin (pg/ml), NEFA (µmol/l), and triglycerides (mmol/l).

## Glucose tolerance tests

Glucose tolerance tests were conducted on *Iap5-1^{full}* and *Iap5-1^{solo}* 16-week-old males (n = 12 per genotype). Following an overnight fast of 15 hr, mice were weighed and baseline blood glucose levels (time 0) were measured using the Glucomen Areo glucometer. A glucose dosage of 2 g/kg was calculated for each mouse based on weight and administered via intraperitoneal injection. Tail blood glucose levels were measured 15, 30, 45, 60, 90, and 120 min post injection. Blood glucose levels were compared between *Iap5-1^{full}* and *Iap5-1^{solo}* mice at each time point using multiple unpaired t-tests corrected for multiple comparisons using the Holm-Šidák method.

## Isolation and storage of biological samples

### Somatic tissues

Liver, cortex, and thymus tissues dissected from adult *Iap5-1^full^* and *Iap5-1^solo^* males were snap frozen in liquid nitrogen and stored at −80°C before use. Ear notches were stored at −20°C.

*GV oocytes.* *Iap5-1^full^* and *Iap5-1^solo^* females were injected interperitoneally with 5 IU of gonadotropin. Ovaries were dissected 40 hr later and placed in M2 medium supplemented with 0.06 g/l potassium penicillin-G, 0.05 g/l streptomycin sulphate, and 240 µM dbcAMP. Ovary follicles were punctures and GV oocytes were detached from their associated cumulus cells using a mouth aspirator and a pulled capillary tube. GV oocytes were stored at −80°C in pools of 100 oocytes collected from multiple females with matching *Iap5-1* genotype.

### Sperm

Mature sperm was isolated from the cauda epididymides of fertile adult *Iap5-1^full^* and *Iap5-1^solo^* males as previously described (*Sharma et al., 2015*) and stored in 1× PBS at −80°C. Sperm samples from different males were analysed separately.

### Blastocysts

*Iap5-1^full^* and *Iap5-1^solo^* females were mated with *Iap5-1^full^* and *Iap5-1^solo^* males, respectively. The uteri and oviducts of pregnant females were dissected 3 days after the identification of a vaginal plug and flushed with M2 medium (Sigma-Aldrich) supplemented with 0.06 g/l potassium penicillin-G and 0.05 g/l streptomycin sulphate. Embryos developed to the blastocyst stage were washed in M2 medium and stored at −80°C. Blastocysts collected from each female were pooled prior to freezing.

### E16.5 embryos and placentas

*Iap5-1^full^* and *Iap5-1^solo^* females were mated with *Iap5-1^full^* and *Iap5-1^solo^* males, respectively. Sixteen days after the identification of a vaginal plug, the uterus was excised from the abdominal cavity and individual E16.5 embryos were removed and rinsed in cold 1× PBS. After removal of the yolk sac and an additional rinse in cold 1× PBS, embryonic tails and full placentas were flash frozen in liquid nitrogen and stored at −80°C.

## DNA extraction

For somatic tissues and ES cells, samples were treated with RNase A at 37°C for 60 min and digested with Proteinase K at 55°C overnight in lysis buffer (10 mM EDTA, 150 mM NaCl, 10 mM Tris-HCl pH 8, 0.1% SDS). gDNA was isolated the next day using a standard phenol–chloroform extraction and ethanol precipitation protocol. The same protocol was followed for sperm gDNA extraction except that the Proteinase K digestion was carried out in equal volumes of Solution A (75 mM NaCl pH 8; 25 mM EDTA) and Solution B (10 mM Tris-HCl pH 8; 10 mM EDTA; 1% SDS; 80 mM DTT) following centrifugation and removal of PBS from thawed sperm. To extract oocyte and blastocyst gDNA, pooled GV oocytes or blastocysts were incubated for 1 hr at 37°C in 14 µl of ddH$_2$O, 1 µl of 1 mg/ml Carrier RNA (QIAGEN), 1 µl of 10% SDS, and 1 µl of 10 mg/ml Proteinase K. After a 15 min incubation at 98°C, samples was directly bisulphite-converted as described below.

## Generation of ESC lines

ESC lines were generated as previously described (*Nichols and Jones, 2017*) with the following modifications. The concentration of MEK inhibitor PDO325901 used in the KSOM+2i and N2B27+2i +LIF media was reduced to 0.2 µM for the initial stages of the protocol and increased to 1 µM following disaggregation of the ICM. The zona pellucide was removed from unhatched embryos using a 10 min pronase digestion at 37°C rather than using acidic Tyrode's solution, and rat serum was used as a source of complement instead of guinea pig serum. Laminin-coated wells were used to ensure proper cell attachment and Accutase solution was used to detach cells prior to passaging. Only male embryos were kept for the generation of ESC lines. Established lines were screened for mycoplasma contamination using the PCR Mycoplasma Test Kit I/C (PromoCell).

## Genotyping

Ear notches were used for *Iap5-1* allelic variant genotyping. Ear notch gDNA was extracted using the PCRBIO Rapid Extract lysis kit (PCR Biosystems) and 1 µl of 1:10 diluted DNA was used as a template for PCR using the REDTaq ReadyMix PCR Reaction Mix (Sigma-Aldrich). The PCR conditions were as follows: (1) 95°C for 4 min 30 s; (2) 94°C for 30 s, optimised T°C for 30 s, 72°C for 30 s, 40 cycles; (3) 72°C for 5 min. For trophectoderm sex-genotyping, trophectoderm lysates were placed in PCR buffer with Proteinase K (50 mM KCl, 10 mM Tris-HCl pH 8.3, 2.5 mM MgCl₂, 0.1 mg/ml gelatin, 0.45% NP40, 0.45% Tween 20, 200 µg/ml Pro K) and incubated at 55°C for 1 hr followed by 95°C for 10 min. Trophectoderm gDNA samples were sex-genotyped by PCR using HotStarTaq DNA Polymerase (QIAGEN) and in the following conditions: (1) 95°C for 3 min; (2) 94°C for 30 s, 56°C for 30 s, 72°C for 55 s, 40 cycles; (3) 72°C for 5 min. Amplified DNA was evaluated by agarose gel electrophoresis. Genotyping primers for *Iap5-1* allelic variants and the Y-linked *Sry* gene are listed in *Supplementary file 1*.

## Sanger sequencing

The PCR primer pairs P1, P2, and P3 were designed to amplify the 5′ half, 3′ half, and the entirety of IAP-*Pgm2* prior to Sanger sequencing, respectively (*Figure 2A*, *Supplementary file 1*). PCR amplification was carried out using the Expand Long Template PCR System (Roche) with the following thermocycler conditions: (1) 94°C for 2 min; (2) 94°C for 10 s, optimised T°C for 30 s, 68°C for 6 min, 10 cycles; (3) 94°C for 15 s, optimised T°C for 30 s, 68°C for 6 min + 20 s each successive cycle, 20 cycles; (4) 68°C for 7 min. Nested PCRs with primer pairs N1, N2, and N3 were performed using Hot-StarTaq DNA Polymerase (QIAGEN) and a 1:20 dilution of the first PCR products as templates (*Figure 2A*, *Supplementary file 1*). Conditions for the nested PCRs were as follows: (1) 95°C for 3 min; (2) 94°C for 30 s, optimised T°C for 30 s, 72°C for 55 s, 40 cycles; (3) 72°C for 5 min. Nested PCR products were purified by gel extraction using the QIAquick Gel Extraction Kit (QIAGEN) according to the manufacturer's instructions and DNA was eluted in ddH₂O. Sanger sequencing was carried out by Source BioScience. Sequencing primers were interspersed regularly across the IAP element (*Supplementary file 1*). Sequence traces were visually examined, and reliable sequences were merged using the EMBOSS *merger* tool. The resulting sequences for the two *Iap5-1* alleles were aligned to the GRCm38/mm10 reference sequence using CLC Sequence Viewer 6. The base-resolution sequence for the *Iap5-1^solo^* allele has been uploaded to GenBank (accession number: MW308129).

## Bisulphite pyrosequencing

Bisulphite conversions were carried out using the two-step modification procedure of the Imprint DNA Modification Kit (Sigma-Aldrich) according to the manufacturer's instructions. 1 µg gDNA was used per conversion with the exception of the oocyte and blastocyst experiments. PyroMark Assay Design SW 2.0 software (QIAGEN) was used to design the pyrosequencing assays (primers listed in *Supplementary file 1*). Target regions were PCR-amplified in technical triplicates from bilsulphite-converted DNA using a biotinylated forward or reverse primer and HotStarTaq DNA Polymerase (QIAGEN). PCR conditions were as follows: (1) 95°C for 3 min; (2) 94°C for 30 s, optimised T°C for 30 s, 72°C for 55 s, 40 cycles; (3) 72°C for 5 min. For low-input pyrosequencing of oocyte and blastocyst DNA, two rounds of PCRs were performed: the first PCR used non-biotinylated primers and 20 amplification cycles (other conditions remained the same); the second PCR used 1 µl of the product from the first PCR as template as well as a biotinylated forward or reverse primer. Following PCR, Streptavidin Sepharose High Performance beads (GE healthcare) were bound to the product in binding buffer (10 mM Tris-HCl pH 7.6, 2 M NaCl, 1 mM EDTA, 0.1% Tween-20) at 1400 rpm for 5 min. The bead-bound biotinylated strands were washed consecutively in 70% ethanol, denaturation solution (0.2 M NaOH), and wash buffer (10 mM Tris-acetate, pH 7.6) using the PyroMark Q96 Vacuum Workstation (QIAGEN). Purified DNA resuspended in annealing buffer (20 mM Tris-acetate pH 7.6, 2 mM magnesium acetate) was incubated with the sequencing primer at 85°C for 4 min. Pyrosequencing was performed on the PyroMark Q96 MD pyrosequencer (QIAGEN) with PyroMark Gold Q96 Reagents and HS Capillary Tips (QIAGEN) according to the manufacturer's instructions. Per cent CpG methylation was calculated by Pyro Q-CpG 1.0.9 software (Biotage) using the ratio of C-to-T at each site. Technical triplicates were averaged, and samples were kept for subsequent analysis if the

standard deviation of technical triplicates did not exceed 5%. Where indicated, methylation levels were averaged across CpGs for each individual at each locus.

## Clonal bisulphite sequencing

Bisulphite-converted liver DNA was amplified by PCR using the primers listed in *Supplementary file 1* and HotStarTaq DNA Polymerase (QIAGEN). PCR conditions were as follows: (1) 95℃ for 10 min; (2) 94℃ for 30 s, 60℃ for 30 s, 72℃ for 30 s, 40 cycles; (3) 72℃ for 5 min. PCR products were purified by gel extraction using the QIAquick Gel Extraction Kit (QIAGEN) and cloned into the pGEM-T Easy vector (Promega) using Stellar Competent Cells (Takara Bio). Sanger sequencing of individual clones was carried out by Source BioScience. QUMA software (*Kumaki et al., 2008*) was used to analyse the sequencing data and generate figures.

## Chromatin immunoprecipitation (ChIP)

ChIP was carried out as previously described with modifications for use on frozen tissue (*Imbeault et al., 2017*). 100 mg of manually powdered frozen liver tissue dissected from adult *Iap5-1$^{full}$* and *Iap5-1$^{solo}$* males was cross-linked in 1% formaldehyde for 10 min at room temperature (RT) and quenched in 250 mM Tris-HCl pH 8 for 10 min at RT on a rotating wheel. Quenched samples were washed twice in 1× PBS supplemented with EDTA-free protease inhibitor cocktail cOmplete (Sigma Aldrich), flash frozen in liquid nitrogen, and stored at −80℃. Fixed liver cells were thawed on ice and sequentially lysed in the following buffers for 10 min at 4℃: LB1 buffer (50 mM HEPES-KOH pH 7.4, 140 mM NaCl, 1 mM EDTA, 0.5 mM EGTA, 10% glycerol, 0.5% NP-40, 0.25% Triton-X-100, 1× EDTA-free cOmpleteTM; one wash), LB2 buffer (10 mM Tris-HCl pH 8.0, 200 mM NaCl, 1 mM EDTA, 0.5 mM EGTA, 1× EDTA-free cOmpleteTM; one wash), and SDS shearing buffer (10 mM Tris-HCl pH 8, 1 mM EDTA, 0.15% SDS, 1× EDTA-free cOmpleteTM; three washes). Samples were centrifuged at 1700 × g for 5 min at 4℃ after every wash. The resulting chromatin was sonicated for eight cycles (one cycle: 30 s on and 30 s off) at 4℃ using a Bioruptor. The sonicated lysate was cleared by centrifugation at maximum speed for 10 min at 4℃ and 10% of the input for each ChIP was stored at −20℃. 50 μl magnetic beads (Protein G Dynabeads, Invitrogen) were pre-blocked in fresh blocking buffer (0.5% BSA in PBS) and incubated with 2.5 μl of polyclonal H3K9me3 antibody (RRID:AB_2532132, Active Motif) or Rabbit IgG (negative control) for 4 hr at 4℃. Antibody-bound beads were washed twice in blocking buffer using a magnetic stand at 4℃. The cleared lysate was topped up to 1 ml SDS shearing buffer + 150 mM NaCl and 1% Triton-X-100 and incubated with the antibody-bound beads on a rotating wheel overnight at 4℃. Non-specifically bound proteins were removed with the following sequential washes at 4℃: low salt buffer (10 mM Tris-HCl pH 8.0, 150 mM NaCl, 1 mM EDTA, 1% Triton X-100, 0.15% SDS, 1 mM PMSF; two washes), high salt buffer (10 mM Tris-HCl pH 8.0, 500 mM NaCl, 1 mM EDTA, 1% Triton X-100, 0.15% SDS, 1 mM PMSF; one wash), LiCl buffer (10 mM Tris-HCl pH 8.0, 1 mM EDTA, 0.5 mM EGTA, 250 mM LiCl, 1% NP40, 1% Na-deoxycholate, 1 mM PMSF; one wash), and 10 mM Tris pH 8.0 (one wash). ChIP samples were resuspended in elution buffer (10 mM Tris pH 8.0, 1 mM EDTA, 1% SDS, 150 mM NaCl), treated with RNaseA at 37℃ for 1 hr at 1100 rpm, and reverse cross-linked overnight at 65℃ at 1100 rpm. The eluted samples were treated with Proteinase K and DNA was purified using the Monarch PCR and DNA Cleanup Kit (NEB).

## RNA extraction and cDNA synthesis

20–30 mg of thymus, liver, cortex, or placenta tissues was homogenised using the MagNA Lyser (Roche) at 6000 × g for 40 s. Thymus and placenta tissues were selected due to the specific expression of *5830416l19Rik* in the thymus and the high expression of *Rell1* and *Pgm2* in the placenta, as reported by NCBI (*Geer et al., 2010*). Liver and cortex were randomly selected to increase the number of tested tissue types. Total RNA was extracted using the AllPrep DNA/RNA Mini Kit (QIAGEN). Tissue DNA was digested on the RNeasy spin column membrane using the RNase-Free DNase Set (QIAGEN) and RNA integrity was confirmed by agarose gel electrophoresis. cDNA was synthesised from 5 μg RNA using random hexamer primers and the RevertAid H Minus First Strand cDNA Synthesis kit (Thermo Scientific).

Quantitative PCR (qPCR) primers were designed using Primer3 software (*Supplementary file 1*). Each reaction was carried out in technical triplicates with Brilliant III Ultra-Fast SYBR Green QPCR

Master Mix (Agilent) on the LightCycler 480 Instrument (Roche) under the following conditions: (1) 95°C for 5 min; (2) 95°C for 10 s, 60°C for 10 s, 72°C for 10 s, 45 cycles; (3) melting curve analysis of 65–95°C. For RT-qPCR, minus RT and no template controls were run for each sample and each primer pair, respectively. Relative expression was normalised to *Hprt1* and *β-actin* expression and calculated using the ΔCt method. For ChIP-qPCR, no template controls and 10% ChIP input were run alongside the H3K9me3 and Rabbit IgG ChIP samples for each primer pair. Enrichment was calculated as per cent input.

## Computational and statistical analyses

ChIP-seq data sets were downloaded in Bigwig format from the GEO database and visualised in the Integrative Genomics Viewer using the NCBI37/mm9 mouse reference genome (*Thorvaldsdóttir et al., 2013*). The three KAP1 ChIP-seq biological replicate tracks were summed in IGV before importing into Adobe Illustrator CC 2020 v24.0 for figure design. GEO accession numbers are listed in *Supplementary file 1*. B6 and CAST DNA sequences at the *Iap5-1* insertion site were extracted from the GRCm38/mm10 and CAST_EiJ_v1 assemblies, respectively, accessed through the UCSC genome browser. The sequence alignment was generated using CLUSTAL OMEGA. All statistical tests in this study were carried out using GraphPad Prism 8 software as indicated in the figure legends.

# Acknowledgements

This work was supported by the Wellcome Trust, United Kingdom (210757/Z/18/Z to ACF-S), the Medical Research Council, United Kingdom (MR/R009791/1 to ACF-S), the Biotechnology and Biological Sciences Research Council, United Kingdom (BB/R009996/1 to ACF-S), and a Cambridge Trust PhD scholarship to TMB. We are grateful to Michael Imbeault, Todd Macfarlan, and Felipe Karam Teixeira for useful discussions, to Amir Hay and Athina Triantou for advice on ChIP, to Rahia Mashoodh for technical assistance during sperm isolation, and to members of the Ferguson-Smith lab for valuable feedback on our work.

# Additional information

### Funding

| Funder | Grant reference number | Author |
| --- | --- | --- |
| Wellcome Trust | 210757/Z/18/Z | Anne C Ferguson-Smith |
| Medical Research Council | MR/R009791/1 | Anne C Ferguson-Smith |
| Biotechnology and Biological Sciences Research Council | BB/R009996/1 | Anne C Ferguson-Smith |
| Cambridge Trust | PhD scholarship | Tessa M Bertozzi |

The funders had no role in study design, data collection and interpretation, or the decision to submit the work for publication.

### Author contributions

Tessa M Bertozzi, Conceptualization, Formal analysis, Investigation, Visualization, Writing - original draft; Nozomi Takahashi, Geula Hanin, Investigation, Writing - review and editing; Anastasiya Kazachenka, Investigation; Anne C Ferguson-Smith, Conceptualization, Supervision, Funding acquisition, Writing - review and editing

### Author ORCIDs

Tessa M Bertozzi (iD) https://orcid.org/0000-0003-2900-6740
Anne C Ferguson-Smith (iD) https://orcid.org/0000-0002-7608-5894

## Ethics

Animal experimentation: Mouse work was carried out in accordance with the Animals (Scientific Procedures) Act 1986 Amendment Regulations 2012 following ethical review by the University of Cambridge Animal Welfare and Ethical Review Body (Home Office project license # PC213320E).

## Decision letter and Author response

Decision letter https://doi.org/10.7554/eLife.65233.sa1
Author response https://doi.org/10.7554/eLife.65233.sa2

## Additional files

### Supplementary files

• Supplementary file 1. Primer sequences and GEO accession numbers.

• Transparent reporting form

### Data availability

Sequencing data have been deposited in GenBank under accession number MW308129. GEO accession codes for publicly available ChIP-seq datasets analysed in this study are listed in Supplementary File 1.

The following dataset was generated:

| Author(s) | Year | Dataset title | Dataset URL | Database and Identifier |
|---|---|---|---|---|
| Bertozzi TM, Hanin G, Takahashi N, Kazachenka A, Ferguson-Smith AC | 2020 | *Mus musculus* domesticus strain C57BL/6J retrotransposon C57iap1_solo, complete sequence | https://www.ncbi.nlm.nih.gov/nuccore/MW308129.1/ | NCBI GenBank, MW308129 |

The following previously published dataset was used:

| Author(s) | Year | Dataset title | Dataset URL | Database and Identifier |
|---|---|---|---|---|
| Wolf G, de Iaco A, Sun MA, Bruno M | 2019 | Retrotransposon reactivation and mobilization upon deletions of megabase-scale KRAB zinc finger gene clusters in mice | https://www.ncbi.nlm.nih.gov/geo/query/acc.cgi?acc=GSE115291 | NCBI Gene Expression Omnibus, GSE115291 |

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
