## [Decision Letter]

**Acceptance summary:**

Retrotransposons can influence genomic function by local chromatin-based position-effects. How much this participates to phenotypic diversity and genome evolution is not well appreciated in mammals. Here the authors identified an IAP retrotransposon copy that exists in two genetic and epigenetic flavors among subpopulations of the mouse C57Bl6/J reference genome. Having one or the other version of this genetically-induced IAP epiallele matters for nearby gene regulation in some tissues and for metabolic features, although the mechanism at play is unknown. This study is impactful for the fields of epigenetics, genomics and mobile DNA.

**Decision letter after peer review:**

Thank you for submitting your article "A spontaneous genetically-induced epiallele at a retrotransposon shapes host genome function" for consideration by *eLife*. Your article has been reviewed by 3 peer reviewers, and the evaluation has been overseen by a Reviewing Editor and Detlef Weigel as the Senior Editor. The following individual involved in review of your submission has agreed to reveal their identity: Geoffrey Faulkner (Reviewer #1).

Summary:

Retrotransposons can influence genomic function by local chromatin-based position-effects, but how much this phenomenon is responsible for generating phenotypic diversity and genome evolution is not well understood in mammals. Here the authors identified an IAP retrotransposon copy that exists in two genetic and epigenetic flavors among subpopulations of the mouse C57Bl6/J reference genome. Having one or the other version of this genetically-induced IAP epiallele matters for nearby gene regulation in some tissues and for metabolic features. This study is impactful for the fields of epigenetics, genomics and mobile DNA.

Essential Revisions:

The authors found that this recent genetic variant occurred by recombination of the two LTRs of an original full length copy, forming a solo LTR version that is unmethylated in somatic tissues, in contrast to other solo LTR elements of the genome. They defined the developmental window when these two IAP alleles diverge in terms of DNA methylation, and involved differential recruitment of sequence-specific KRAB-ZFP as responsible for the different epigenetic states. Although this is a one-case study, this work unambiguously shows that genetic changes in IAP copies influence their own epigenetic states, in an autonomous manner (independently of the genomic environment). Finally, this work represents an important piece of warning of the unsuspected genetic/epigenetic diversity of a very well-characterized, inbred mouse strain, the reference C57Bl6/J genome, and may provide explanations for discrepant results and/or phenotypes obtained among different C57Bl6/J mouse cohorts.

The reviewers unanimously recognized the originally of the findings, the quality of the experimental approaches (with notably strong mouse genetics with reciprocal crosses and backcrosses between mouse strains) and the accurate interpretation of the data.

A limited points for improvements were suggested.

1. The current bisulfite sequencing approach analyses the first/most 5' four CpGs found in the 5' LTR of C57iap1^solo^ and C57iap1^full^. In Figure 1A, the authors nicely check that the four last/most 3' CpGs in the 3' LTR are similarly methylated to the first 4 CpGs of the 5' LTR of the C57iap1^full^. However, it is unclear whether the methylation status of these 4 CpGs is reflective of the methylation across the LTRs. Moreover, it is unknown as to whether the 5' end of the C57iap1^solo^ originates from the 5' LTR or the 3'LTR. If the latter, C57iap1^solo^ could appear less methylated than C57iap1^full^ because the approach is measuring different CpGs. The 5' end of the C57iap1^solo^ should then be compared with the 5'end of the 3'LTR of the C57iap1^full^. Do the authors have data to exclude this possibility? Bisulfite pyrosequencing does not allow analyzing large amplicons. But bisulfite cloning sequencing could be attempted (on one somatic tissue) to investigate larger parts of the 5'LTR and the 3'LTR of the C57iap1^full^ and larger part of the LTR of the C57iap1^solo^.

2. The reviewers suggest mining into public data, and more particularly long-read sequencing data, to try to identify other cases of conversion of full length IAPs in the reference C57Bl6/J genome into solo LTRs in sub-C57BL6/J populations. This would document how pervasive this could be and how much it may contribute to variable genome regulation in this reference mouse strain. But the reviewers acknowledge that it is unclear as to whether such datasets exist.

3. The authors were careful stating that the C57iap1 polymorphism is "associated" with transcription of nearby genes, as they did not investigate as to how the C57iap1^solo^ regulates these nearby genes, acting as an alternative promoter or an enhancer. Abundant transcriptomic data exist and may resolve this easily, provided that some of them emanate from the C57bl6/J subpopulation with the C57iap1^solo^ polymorphism. If not, RT-PCR approaches may be attempted.

---

## [Author Response]

Essential Revisions:The authors found that this recent genetic variant occurred by recombination of the two LTRs of an original full length copy, forming a solo LTR version that is unmethylated in somatic tissues, in contrast to other solo LTR elements of the genome. They defined the developmental window when these two IAP alleles diverge in terms of DNA methylation, and involved differential recruitment of sequence-specific KRAB-ZFP as responsible for the different epigenetic states. Although this is a one-case study, this work unambiguously shows that genetic changes in IAP copies influence their own epigenetic states, in an autonomous manner (independently of the genomic environment). Finally, this work represents an important piece of warning of the unsuspected genetic/epigenetic diversity of a very well-characterized, inbred mouse strain, the reference C57Bl6/J genome, and may provide explanations for discrepant results and/or phenotypes obtained among different C57Bl6/J mouse cohorts.The reviewers unanimously recognized the originally of the findings, the quality of the experimental approaches (with notably strong mouse genetics with reciprocal crosses and backcrosses between mouse strains) and the accurate interpretation of the data.A limited points for improvements were suggested.1. The current bisulfite sequencing approach analyses the first/most 5' four CpGs found in the 5' LTR of C57iap1^solo^ and C57iap1^full^. In Figure 1A, the authors nicely check that the four last/most 3' CpGs in the 3' LTR are similarly methylated to the first 4 CpGs of the 5' LTR of the C57iap1^full^. However, it is unclear whether the methylation status of these 4 CpGs is reflective of the methylation across the LTRs. Moreover, it is unknown as to whether the 5' end of the C57iap1^solo^ originates from the 5' LTR or the 3'LTR. If the latter, C57iap1^solo^ could appear less methylated than C57iap1^full^ because the approach is measuring different CpGs. The 5' end of the C57iap1^solo^ should then be compared with the 5'end of the 3'LTR of the C57iap1^full^. Do the authors have data to exclude this possibility? Bisulfite pyrosequencing does not allow analyzing large amplicons. But bisulfite cloning sequencing could be attempted (on one somatic tissue) to investigate larger parts of the 5'LTR and the 3'LTR of the C57iap1^full^ and larger part of the LTR of the C57iap1^solo^.

We carried out the suggested clonal bisulphite sequencing experiment and managed to assess all of the CpGs in the 5' and 3' LTRs of *C57iap1^full^* (*Iap5-1^full^* in our revised version) as well as all of the CpGs in the *C57iap1^solo^* (*Iap5-1^solo^* in our revised version) LTR. These data confirm that DNA methylation of the most distal CpGs quantified throughout the manuscript using bisulphite pyrosequencing are indeed reflective of DNA methylation across the entire LTRs. They also exclude the possibility that the difference in DNA methylation between *Iap5-1^full^* and *Iap5-1^solo^* are due to the quantification of different CpGs. In addition, we were able to measure the methylation state of a few CpGs located inside the internal region proximal to the 5' LTR of *Iap5-1^full^*, suggesting that the entire *Iap5-1^full^* provirus is heavily methylated. These results have been included in the updated manuscript in Figure 2—supplement figure 2 and are discussed in the text (lines 186-189).

2. The reviewers suggest mining into public data, and more particularly long-read sequencing data, to try to identify other cases of conversion of full length IAPs in the reference C57Bl6/J genome into solo LTRs in sub-C57BL6/J populations. This would document how pervasive this could be and how much it may contribute to variable genome regulation in this reference mouse strain. But the reviewers acknowledge that it is unclear as to whether such datasets exist.

Whole-genome long-read sequencing datasets generated from C57BL/6J genomic DNA are indeed starting to appear on online repositories, opening up new opportunities to investigate naturally occurring genetic variation in commonly used inbred mouse strains. However, in addition to building a robust computation pipeline to identify new solo LTRs (as well as other transposable element-derived polymorphisms), we are aiming to validate the findings experimentally on various C56BL/6J sub-populations and conduct secondary analyses on the genomic distribution, epigenetic state, and functional consequences of the identified polymorphisms. This interesting future direction is therefore outside the scope of this manuscript.

3. The authors were careful stating that the C57iap1 polymorphism is "associated" with transcription of nearby genes, as they did not investigate as to how the C57iap1^solo^ regulates these nearby genes, acting as an alternative promoter or an enhancer. Abundant transcriptomic data exist and may resolve this easily, provided that some of them emanate from the C57bl6/J subpopulation with the C57iap1^solo^ polymorphism. If not, RT-PCR approaches may be attempted.

We did not detect transcripts initiating from the *Iap5-1^solo^* LTR in publicly available transcriptomic data but we are hesitant to make conclusions based on this given that it is impossible to determine which *Iap5-1* variant was present in the C57BL/6J samples used to generate the datasets. As suggested by the reviewers, we carried out additional RT-qPCR experiments and found that the unique regions bordering *Iap5-1^solo^* are transcribed while those bordering *Iap5-1^full^* are not. We have included these results in Figure 5—supplement figure 1 and have added a sentence describing them in the text (lines 343-345). However, we do not believe that these data conclusively indicate that *Iap5-1^solo^* is acting as an alternative promoter and have therefore refrained from interpreting them in this manner.